# Update on Anticoagulation Strategies in Patients with ECMO—A Narrative Review

**DOI:** 10.3390/jcm12186067

**Published:** 2023-09-20

**Authors:** Ján Šoltés, Michal Skribuckij, Hynek Říha, Michal Lipš, Pavel Michálek, Martin Balík, Michal Pořízka

**Affiliations:** 1Department of Anaesthesiology and Intensive Care Medicine, First Faculty of Medicine, Charles University in Prague and General University Hospital in Prague, 12808 Prague, Czech Republic; jan.soltes@vfn.cz (J.Š.); dr.hynek.riha@gmail.com (H.Ř.); michal.lips@vfn.cz (M.L.); pavel.michalek@vfn.cz (P.M.); martin.balik@vfn.cz (M.B.); 2Emergency Service of Central Bohemia, Vančurova 1544, 27201 Kladno, Czech Republic; 3Department of Anaesthesia, Golden Jubilee University National Hospital, Clydebank G81 4DY, UK; michal.skribu@gmail.com; 4Department of Anaesthesiology and Intensive Care Medicine, Institute for Clinical and Experimental Medicine, 14021 Prague, Czech Republic; 5Department of Anaesthesia, Antrim Area Hospital, Antrim BT41 2RL, UK

**Keywords:** ECMO, anticoagulation, heparin, low-molecular-weight heparin, COVID, coagulation monitoring, anticoagulation target

## Abstract

The use of extracorporeal membrane oxygenation (ECMO) has recently increased exponentially. ECMO has become the preferred mode of organ support in refractory respiratory or circulatory failure. The fragile balance of haemostasis physiology is massively altered by the patient’s critical condition and specifically the aetiology of the underlying disease. Furthermore, an application of ECMO conveys another disturbance of haemostasis due to blood-circuit interaction and the presence of an oxygenator. The purpose of this review is to summarise current knowledge on the anticoagulation management in patients undergoing ECMO therapy. The unfractionated heparin modality with monitoring of activated partial thromboplastin tests is considered to be a gold standard for anticoagulation in this specific subgroup of intensive care patients. However, alternative modalities with other agents are comprehensively discussed. Furthermore, other ways of monitoring can represent the actual state of coagulation in a more complex fashion, such as thromboelastometric/graphic methods, and might become more frequent. In conclusion, the coagulation system of patients with ECMO is altered by multiple variables, and there is a significant lack of evidence in this area. Therefore, a highly individualised approach is the best solution today.

## 1. Introduction

The use of extracorporeal membrane oxygenation (ECMO) has increased exponentially in recent times [1], particularly following the H1N1 and COVID-19 pandemics [2]. It represents a modified cardiopulmonary bypass system that facilitates extracorporeal decarboxylation and oxygenation. The application of ECMO encompasses various scenarios involving refractory both heart and lung failure, and these indications have been established for adult and paediatric patients over recent decades, bringing a chance to improve the outcome of these patients [3]. The essential components of the ECMO circuit include a pump, a membrane oxygenator, a heat exchanger, a venous cannula, an arterial or venous infusion cannula, tubing, and connectors. It can be configured either in a venovenous (VV) mode, which is employed for the management of severe respiratory failure, or in a venoarterial (VA) mode, which provides simultaneous support for both respiratory and cardiac functions [4]. VV ECMO can be initiated using peripheral approaches involving two cannulae (femorojugular or femorofemoral) or it can be established using a double-lumen cannula inserted under ultrasound and echocardiographic guidance through the internal jugular vein. Alternatively, the drainage cannula is introduced through a venous access point in VA ECMO, while the return cannula is placed into an arterial access site. VA ECMO classification can be categorised as either central or peripheral, depending on the specific vessels used for cannulation [5]. In the central configuration, the drainage cannula can be directly inserted into the right atrium, with the return cannula placed in the ascending segment of the aorta. In the peripheral configuration, blood can be extracted through the femoral or jugular veins, and it is then returned to the patient through the carotid, axillary, or femoral arteries [5].

As anticipated, ECMO complications are prevalent and linked to a substantial rise in morbidity and mortality [3]. The risks associated with the ECMO procedure can be categorised into two main groups: mechanical and patient-related medical complications. Mechanical complications are represented by issues such as oxygenator or pump failure, heat exchanger malfunction, cannula-related problems, and disruptions in the circuit. On the other hand, patient-related medical complications include bleeding, thromboembolic events, neurological issues, the development of additional organ failures (cardiovascular, renal, or hepatic), metabolic disorders, and infections [3,5]. Thus, it is important to note that despite recent advancements, the mortality rates associated with ECMO use remain high for both modalities [6,7].

## 2. ECMO Circuit Interactions with Coagulation System

A fragile balance of haemostasis physiology is massively altered in a critical patient, and specifically by the aetiology of the primary disease. Furthermore, the use of ECMO brings another disturbance to haemostasis related to blood interaction with the circuit and an oxygenator [8]. In the extracorporeal circuit, blood is exposed to a non-biologic surface. This inevitably leads to up-regulation of pro-inflammatory and procoagulant pathways. The pathophysiological background of this process is based on the activation of coagulation factor XII and platelet activation with their subsequent dysfunction [9]. These are the reasons why, in patients with ECMO, it is necessary to balance between possible thrombotic and bleeding complications, which are unfortunately still very frequent. According to the ELSO register, thrombotic complications are more common than bleeding, but bleeding carries a significantly higher mortality risk [10]. Furthermore, under specific conditions, for example, severe bleeding or emergency surgery, anticoagulation is usually temporarily suspended. Surprisingly, a recent systematic review analysing anticoagulant-free ECMO support found no increased risk of thrombotic or bleeding complications compared to systemic anticoagulation [8].

Although the updated ELSO guidelines (2021) were published recently, the rapid development in the field and a lack of evidence-based recommendations, as conceded in the guidelines [11], justify a need for a review. According to the current literature, several pharmacological agents can be used as anticoagulants in patients with ECMO. These are: unfractionated heparin (UFH), low-molecular-weight heparin (LMWH), fondaparinux, and direct thrombin inhibitors [11,12]. Nevertheless, more than choosing the right agent, another challenge is finding and deciding on the appropriate target and the most suitable testing method. First, standard coagulation tests (prothrombin time/INR and aPTT) are often altered by the patient’s critical condition (especially shock, sepsis and liver dysfunction). Thus, coagulation times might be spontaneously prolonged. At the same time, testing the patient who shows prolonged coagulation times with viscoelastometric tests (VETs) could demonstrate a hypercoagulation state. Another major issue in patients with ECMO is the occurrence of low platelet count, making the topic even more complex.

In this narrative review, we summarise the knowledge on choosing an anticoagulant agent with an emphasis on the most commonly used UFH. Another aspect is the appropriate method of testing, and the last highlighted issue is the gentle balance between thrombotic and bleeding complications.

## 3. Considerations for VV versus VA ECMO

The anticoagulation strategy may differ between the VV and VA modalities. The VV ECMO may tend to have more bleeding complications at the insertion [10], likely related to a rapid decrease in pCO_2_ together with a bolus anticoagulation with cannulation [13]. The VA ECMO may tend more towards thrombosis-related complications, especially with poor LV unloading and intracardiac thrombus formation, and during weaning of the VA modality due to low blood flow (down to 0.5 L/min) in the extracorporeal circuit and oxygenator [10]. Both modalities are fraught with cannulation-related deep venous thrombosis, i.e., related to the drainage cannulas and the return cannula of the VV ECMO [14]. The low-flow segment of the peripheral VA ECMO is between the return arterial cannula and the prograde superficial femoral artery cannula. Here, the arterial thrombus formation indicates a surgical approach with femoral thrombectomy, rather than simple bedside extraction with compression or a closure device.

## 4. Considerations for Exhaustion of the Coagulation System

Exhaustion of the coagulation cascade typically results from a procoagulant state. A systemic inflammatory syndrome involves major triggers and regulators of the coagulation system. These are damaged endothelium by inflammation, activated intrinsic system with alteration in the protein C pathway, activated platelets, thromboplastin and activation of the extrinsic system, and finally involving the changes in fibrinolysis [15]. A typical example of coagulation activation is a sepsis-related disseminated inflammatory coagulopathy. Here, the mild prolongation of classic coagulation monitors like APTT, PT are accompanied by an increase in acute-phase proteins, including fibrinogen, and a rise in fibrin degradation products (D-dimers) with a D-dimer-related prolongation of the thrombin time [15,16,17]. An activation of platelets may be seen in the aggregometry tests or in the VETs. Specific tests may even show increased activity of certain factors like vWf. With progression of the systemic inflammation and exhaustion of the coagulation cascade, the thrombin activity decreases, accompanied by further extension of the coagulation times (APTT, PT, TT), decreasing fibrinogen, vWf, antithrombin III, and platelet count. The fibrinolytic pathway may require testing with the VETs to reveal hyper- or hypoactivity of plasmin, with potential implications for therapy [18].

## 5. Special Considerations for COVID-19 and ECMO

Since the outbreak of SARS-CoV-2, an increased risk of thrombosis has been reported in patients with COVID-19 [19]. COVID-19-associated coagulopathy (CAC) manifests as micro- or macrothrombi formation, which can cause damage to multiple organs (lungs, heart, brain, kidneys) [19], resulting in an increase in morbidity and mortality. Compared to other viral infections, patients with COVID-19 have higher rates and severity of clotting events, which is underlined by laboratory findings of elevated plasma levels of D-dimer, C-reactive protein, fibrinogen and P-selectin [19]. Pathophysiological background includes a dysregulation among the inflammatory, immune, coagulation, fibrinolytic, complement and kallikrein–kinin systems. Thus, the term “immune thrombosis” has been established [19].

VV ECMO has been extensively used in the treatment of COVID-19-related ARDS. Less frequently, ECMO in VA configuration has been indicated for combined cardiac-respiratory failure [20]. Multiple studies showed that COVID-19 is connected to an increased risk of endothelial inflammation-mediated thrombosis [21,22]. As a result, the therapeutic anticoagulation instead of a prophylactic modality has been accepted for critically ill COVID-19 patients [16]. However, this approach showed serious limitations for patients requiring ECMO support. Mansour et al. published a nationwide trial involving 620 patients with COVID-19 and ECMO support. The study demonstrated a high prevalence of all bleeding events (49.4%) regardless of their severity.

A different methodology including only life-threatening bleeds, like in the ELSO registry or the International Society on Thrombosis and Haemostasis registry, may not elucidate the difference against the control population, because the bleeding events were independently associated with a higher in-hospital mortality on day 90. In addition, the rate of thrombotic complications was also high (36%), but thrombotic events were not associated with higher mortality rates [21]. These results differ from the large multicenter studies in the population of ECMO patients without COVID-19, which showed a significant increase in mortality due to thrombosis. It is essential to highlight that in the Mansour et al. study, intracranial bleeding (ICH) was reported in an alarming 8% of patients [19]. This finding supports multiple recent publications [20], suggesting a higher risk of intracranial bleeding in COVID-19 patients treated with ECMO [22]. Furthermore, the systematic review and meta-analysis of Jin et al. [23] included 23 studies with 6878 subjects. The results showed that a major bleeding event occurred in 37.4% of cases. ICH was reported in 9.9% of the subjects. The increased risk of ICH was shown in COVID-19 ECMO patients compared to the population without COVID-19 (RR = 2.23 (95% CI: 1.32–3.75)). Unrestricted anticoagulation regimens recommended early in the COVID-19 pandemic [24] may explain the increased incidence of significant bleeding events in patients with ECMO. The key to understanding the change from a hypercoagulable status of severe COVID-19 viral sepsis to a hypocoagulating frequently bleeding COVID-19 patient on ECMO is the extensive endothelial inflammation caused by the coronavirus [15]. COVID-19-caused inflammation of the endothelium initially upregulates the activity of von Willebrand factor (vWf) and factor VIII, leading to inadequate balance by the otherwise often normal levels of its depolymerase (ADAMTS-13). The prothrombotic state associates with high thrombin activity due to contact activation of the coagulation cascade and high levels of the acute-phase inflammatory protein fibrinogen [25]. With an insertion of the ECMO circuit, the situation changes. Acquired vWf disease accompanies almost uniformly current ECMO circuits and is even more relevant to severe COVID-19 patients on ECMO. With the commencement of the ECMO therapy, the upregulated vWf decreases [26]. This, together with an exhaustion of the coagulation system and low thrombin activity and a decrease in the initially upregulated fibrinogen levels, may create a dangerous cumulation of factors leading to life-threatening bleeding [17,27]. Moreover, the alterations in the fibrinolytic system are difficult to predict and should be assessed individually [18]. The endothelial inflammation of the haematoencephalic barrier may potentially relate to the rates of ICH [28]. The potential role of rapid change in pCO_2_ during ECMO initiation on the incidence of ICH has already been discussed as a potential causative factor for an ICH.

Currently, there is no consensus on an intensified anticoagulation practice for this particular patient population or those on only prophylactic anticoagulation, because the reported numbers of serious thrombotic events also related to ECMO are not negligible [29]. The increased prevalence of ICH highlights the need for a strictly controlled and individualised anticoagulation modality.

## 6. Unfractionated Heparin as Gold Standard

### 6.1. Introduction to Heparin

In contrast to most pharmacologic agents used in medicine, heparin is a heterogeneous compound consisting of molecules with variable molecular weights. The molecular weight of majority of the chains varies from 15,000 to 19,000 Daltons. This heterogeneity of heparin also leads to variable biologic effects. Therefore, the dose is usually not given in milligrams but in standardised international units—IU [30,31].

Heparin is used extensively because its effect can be easily titrated, the half-life is short, and its antidote (protamine) is widely accessible [32]. At its effect site, a bond with antithrombin (AT; previously antithrombin III) is created by the pentasaccharide segment of the molecule. The effect of AT is 1000 times multiplied when heparin is bound to its molecule. AT is a potent inhibitor of thrombin (factor II), factor Ixa, and Xa. Subsequently, a thrombin-dependent platelet activation and factors V, VIII, and XI are inhibited [30,33]. In conclusion, the mechanism of action of heparin is complex.

A clinician may face some typical situations when administering a heparin anticoagulation. First, an excessive dose of UFH may be necessary to maintain the desired target of anticoagulation. Second is the occurrence of unexpected and sometime severe low platelet count in the so-called heparin-induced thrombocytopenia (HIT), and lastly in deciding about the right method of monitoring and choosing the right target with a lack of evidence.

### 6.2. Monitoring and Target

Although UFH is the most frequently used anticoagulant in this specific group of patients, there are no detailed recommendations and a significant lack of data in scientific databases. According to the 2021 ELSO Adult and Paediatric Anticoagulation Guidelines [11], the monitoring of anticoagulation by ACT (in the range of 180–200 s [34]; with a low-concentration cartridge [35]) is considered a gold standard for the guidance of anticoagulation therapy in patients with ECMO. The widely spread standard laboratory-activated partial thromboplastin test (aPTT) is—owing to the guidelines—a gold standard assay for monitoring UFH. However, the laboratory aPTT assay may be influenced by multiple factors compared to the anti-Xa test, and recent data suggest purchasing both aPTT and the anti-Xa tests in parallel, especially in ECMO patients. This approach may allow for a calibration of the aPTT and prevent underdosing or overdosing with the unfractionated heparin modality [36]. Bedside testing for ACT and aPTT is possible, which might differ from standardised laboratory aPTT assays. There is evidence that decisions in anticoagulation management are different for a laboratory-obtained aPTT compared with bedside testing of ACT and aPTT. Furthermore, there is older evidence that standard aPTT from the laboratory showed a stronger correlation with plasma UFH concentration than the mentioned bedside assays [37]. According to current research, other authors prefer not to use ACT [34].

ELSO suggested repeating the test every 1–2 h in case of ACT and every 6–12 h for monitoring the anticoagulation effect with aPTT. Great emphasis is placed on individual protocols of particular ECMO centres, which depend on local experiences. In the case of resistance to heparin, an examination with anti-Xa might be helpful. Routine supplementation with AT does not have scientific evidence.

Under special circumstances, that is, spontaneously prolonged aPTT, VETs might give a reliable image of the actual coagulation status. Thus, hypercoagulability can be uncovered, and anticoagulation can be initiated regardless of spontaneously prolonged aPTT. In patients supported on ECMO, multiple studies evaluated the safety and feasibility of a VET-driven strategy to titrate heparin versus the “conventional” standard of care based on aPTT monitoring [34]. The improved trend of less bleeding complications was observed in a TEG^®^-guided group (target 16–24 min of the R parameter). Furthermore, required doses of heparin were lower in a TEG^®^ group than aPTT-guided care [38]. Another study investigated optimal values for citrated kaolin TEG R^®^ time. They concluded that TEG^®^ R time > 17.85 min (sensitivity 84%, specificity 68%, PPV 82%, and NPV 59%) is able to minimise the risk of thrombosis in paediatric and neonatal ECMO patients [39]. The essential issue is that according to ELSO [11], there are no relevant data to guide current practice. A whole paradigm about targeting aPTT ratio between 1.5 and 2.5 is based on one prospective non-ECMO study from 1972. The trial concluded that keeping aPTT in this range reduces the prevalence of recurrent venous thromboembolism. Another historic study conducted on patients undergoing continuous renal replacement therapy from 1996 claims that an aPTT ratio of 1.5 can cut the risk of circuit thrombosis by half. On the other hand, the risk of bleeding complications is threefold increased [40]. This lack of crucial data explains why many recommendations exist for an individualised approach.

The most appropriate current approach is a highly individually titrated anticoagulation target. Multiple variables such as platelet count, fibrinogen level, initial results of INR, aPTT, VETs, evidence of prothrombotic/haemorrhagic condition, and clinical signs of thrombosis or bleeding should be considered before making the optimal decision.

The rationale behind this precise anticoagulation titration is to keep the fragile balance between the two extreme situations. While bleeding and thrombosis remain common complications in ECMO patients, haemorrhagic events substantially influence mortality (emphasising fatal intra-cranial bleeding) [10,41]. According to the systematic review of Abruzzo et al., early computed tomography (CT) diagnosis of deep venous thrombosis was obtained in a number of patients as high as 71.4%, and pulmonary embolism was observed in 16.2% of the patients [14]. Therefore, a routine check-up for thrombosis should be considered after ECMO decannulation. Ultrasound examination seems like an ideal non-invasive method without ionising radiation, but unfortunately has a higher rate of false-negative results compared to [14].

Furthermore, not only the target of anticoagulation is what can increase the risk of adverse bleeding. The extensive ELSO data show that rapid changes in PCO_2_ may also independently increase the risk of neurological complications (seizures, intracranial bleeding) [13]. That is why a gentle approach should be applied to restore blood gas homeostasis without rapid changes.

### 6.3. Heparin Resistance

This condition is defined as the need for an excessive dose of UFH to achieve the desired laboratory result. For some, this is defined as a dose >35,000 IU/24 h required to achieve a subtherapeutic range. One of the reasons for heparin resistance is considered to be AT deficiency [42]. Furthermore, other risk factors increase the risk of heparin resistance, including smoking, chronic obstructive pulmonary disease, liver dysfunction, nephropathies, hypoalbuminaemia, thrombocytosis, intra-aortic balloon pump, prolonged UFH treatment and presence of an ECMO circuit. AT deficiency might be acquired or (rarely) congenital [33,42,43]. According to some sources, a very high prevalence of heparin resistance was observed in COVID-19 patients (up to 62%) [44]. To demonstrate that this is not a simple one-way pathophysiological pathway, it is necessary to mention that low AT level does not automatically cause heparin resistance. On the other hand, a normal AT level might be present in cases of resistance to heparin. Exogenous AT might look like a simple solution, but it is not widely consensually recommended and might be a reason for adverse bleeding complications.

Because of the heterogeneous nature of UFH molecule size, molecules of lower molecular weight may be more prevalent in some drugs. Hence, in cases of apparent heparin resistance, anti-Xa monitoring might be more appropriate and explain inadequate response in aPTT [30]. In this case, following up with a targeted anti-Xa approach should be considered.

## 7. Low-Molecular-Weight Heparins and Fondaparinux

Low-molecular-weight heparins (LMWH) are produced by depolymerisation of UFH to approximately one-third of the original size of the molecule (4000–6500 D), thus creating a shorter saccharide chain. Similarly to UFH, there is considerable variability in molecular weight and chemical properties [45].

Fondaparinux is a fully synthetic pentasaccharide, and its units are structurally similar to the cleaved monomeric units of UFH [46]. It was developed during an effort to solve problems associated with naturally sourced UFH (e.g., variability in composition, potential contamination) and eliminate side effects such as HIT [47]. Although LMWH has also been associated with HIT, the risk is considerably lower than with UFH and negligible in the case of fondaparinux [48].

Similarly to UFH, LMWH acts by activating AT. However, due to its relatively short molecular chain, LMWH cannot form the tertiary complex with AT and thrombin, which is essential for the inactivation of thrombin. In contrast, factor Xa is inactivated by AT without the need to form the tertiary complex with thrombin. Thus, in the presence of LMWH, AT exhibits preferential binding to factor Xa [45,49]. Due to its chemical structure, fondaparinux targets the same pathway as LMWH, which means binding to AT with following deactivation of Xa [47].

Both LMWH and fondaparinux are eliminated by the kidneys, have a longer plasma half-life, better bioavailability, and in a standard setting, a more predictable dose response than UFH. However, they cannot be easily reversed by an antidote and their effect is prolonged in patients with renal failure [45,50]. Another disadvantage of standard subcutaneously administered LMWH is its variable bioavailability. This is significantly influenced by peripheral perfusion, often affected by the use of vasopressor medications [51,52]. Therefore, the LMWH administration should always be by the intravenous route.

### 7.1. Monitoring and Target

#### 7.1.1. LMWH

The primary method of monitoring both LMWH and fondaparinux is by measuring the anti-Xa levels. The effect of fondaparinux is best measured using a specifically calibrated anti-Xa assay because the standard LMWH-calibrated assay overestimates the concentration of fondaparinux by about 20% [39]. Current recommendations for monitoring and anticoagulation targets in ECMO patients are outlined in the 2021 ELSO Adult and Pediatric anticoagulation guidelines. The guidelines suggest a target anti-Xa, albeit only in the context of UFH modality [11].

Concerning LMWH, published articles suggest that most centres opt for prophylactic dosage, especially in V-V ECMO [53,54]. This is in agreement with the current trend towards less or no anticoagulation for the V-V ECMO, supported mainly by retrospective studies and only one prospective pilot study [55]. Therefore, due to the lack of robust prospective data, the prophylactic approach is not included in the 2021 ELSO guidelines.

LMWHs have gradually replaced UFH in many medical indications, and their superiority in prophylactic anticoagulation in critically ill patients has been consistently described [56,57,58]. Additionally, LMWHs have been used in prophylaxis in other types of extracorporeal circuits, for example, continuous renal replacement therapy (CRRT) [59]. Some data originating from a non-intensive care environment even suggest their possible superiority in therapeutic anticoagulation [60,61]. Despite this, there is a paucity of evidence on its use in patients with ECMO. This is also reflected in the 2021 ELSO guidelines, where the emphasis on using UFH/direct thrombin inhibitors is prominent [62].

A single-centre observational study evaluated administering a prophylactic dose of enoxaparin to more than 60 patients with nonsurgical VV ECMO. It showed fewer bleeding events with a higher incidence of thrombotic complications than studies with therapeutic anticoagulation with UFH [53]. Regarding surgical patients, one single-centre retrospective study investigated prophylactic subcutaneous enoxaparin versus intravenous UFH used in perioperative ECMO in 102 lung transplant patients and showed no difference in bleeding risk and fewer thrombotic episodes in the LMWH group [63].

The use of VETs for monitoring of the “LMWH” effect have not been standardised yet, and thus it is not suitable for titration of anticoagulant effect in fragile ECMO patients [64].

#### 7.1.2. Fondaparinux

Similarly to LMWH, the evidence for using fondaparinux in patients with ECMO is scarce, albeit the published articles generally describe positive outcomes. Only several case reports and series on the topic were published, all with one common theme: patients on ECMO who developed HIT. The prophylactic dose once daily appears to be this group’s most prevalent anticoagulation strategy [65,66,67]. In most published case reports, only a prophylactic dose of fondaparinux was used, and anti-Xa was not measured [65,66]. In one case series published by Rychlíčková et al. [67], the therapeutic levels were targeted, the anti-Xa levels were measured, and the dosages were adjusted accordingly. The prophylactic approach using fondaparinux could cause a problem because the low anticoagulation target might not be sufficient for patients with HIT, as suggested by the 2018 American Society of Hematology guideline for management of heparin-induced thrombocytopenia. This guideline recommends achieving therapeutic-intensity anticoagulation [62]. Although this target has not been established for patients with ECMO, it would seem reasonable that higher levels of anticoagulation should be targeted in patients with HIT. Management of anticoagulation with fondaparinux based on VETs is not appropriate. According to minimal data, VETs are only able to detect supratherapeutic doses of fondaparinux [64].

In summary, anticoagulation with fondaparinux and LMWH in patients with ECMO is challenging and guidelines for the optimal target have yet to be established. An individual approach to each case is essential and test results must be interpreted in the context of both the patient and the circuit.

### 7.2. Heparin-Induced Thrombocytopenia

A decrease in platelet count is a common complication of ECMO therapy (both in VV and VA) and occurs in up to 50% of subjects with ECMO. Post-cardiotomy ECMO patients represent a group with a higher risk of developing significant thrombocytopenia. There are many possibilities for the aetiology of thrombocytopenia in this particular group of patients: the foreign surface of the circuit, platelet activation, nonspecific activation of the inflammatory cascade, sepsis, medications (immunosuppressives, PDE III inhibitors), surgery, and bleeding. Often it is not possible to highlight one particular reason. Thus, aetiology is frequently considered multifactorial [68,69,70].

The prevalence of HIT in patients undergoing ECMO therapy ranges from 0.5% to 5.0% [70,71]. These numbers are probably underestimated, as HIT testing is not routinely performed in all patients. HIT is an antibody-mediated side effect of UFH administration that is characterised by the onset of thrombocytopenia, usually 5–10 days after the beginning of therapy. Paradoxically, the result of HIT is a prothrombotic state with a high risk of thrombosis, especially in cases of rare HIT II. When the thrombocytopenia is severe (<20 × 10^9^/L), then a risk of bleeding arises [68,70,71].

Typically, PF4 is stored in α-platelet granules (and released after adequate activation). The physiological function of PF4 is in binding to heparan sulphate, an endogenous substance very similar to UFH. However, its ability to bind with UFH shows a much higher affinity. Due to the newly created complex heparin–PF4 with the Fc site of the antibodies, the platelets and immune cells are activated. This results in thrombin activation and scavenging of platelets by macrophages [71,72].

The correct HIT diagnostic algorithm should be based on three positive criteria [72]:(1)UFH (or LMWH) exposure.(2)At least one clinical or laboratory result (low platelet count or new venous/arterial thrombosis).(3)Evidence of specific HIT antibodies.

HIT-specific antibodies are examined by ELISA for the detection of the heparin–PF4 complex. This test shows a negative predictive value of 98–99%. This means that the negative result is very reliable for excluding the possibility of HIT. However, the positive predictive value is only 2–15%. Clinical suspicion and mentioned screening methods above predict the result of the confirmation methods well, so together they might often be sufficient. Confirmation tests with higher specificity are based on examination of platelet activation, e.g., SRA (serotonin-release assay) or HIPA (heparin-induced platelet activation assay) [73].

The treatment of HIT is based on discontinuing UFH administration and switching to another anticoagulation modality. Direct thrombin inhibitors, such as argatroban or bivalirudin are recommended [72]. The switch to fondaparinux is also possible and widely accepted as a drug of choice [73].

The risk of developing HIT when using LMWH is lower than UFH, but remains a genuine concern. The pathophysiological mechanisms of HIT may differ between UFH and LMWH, primarily due to the smaller size of LMWH molecules. The size of the heparin molecule plays a crucial role in its capacity to bind with PF4, as confirmed by experimental studies. This suggests that LMWH is less likely to provoke HIT when compared to UFH, as research indicates a higher occurrence of anti-H–PF4 antibody formation in patients treated with UFH compared to those receiving LMWH [48]. Nevertheless, it is essential to acknowledge that LMWH can still potentially induce HIT because specific LMWH preparations may contain larger molecules that can interact with PF4 [48].

## 8. Direct Thrombin Inhibitors

The direct thrombin inhibitors (DTIs) represent short-acting anticoagulants with a mechanism of action that primarily inhibits thrombin, a key enzyme involved in the coagulation cascade. By targeting both soluble and fibrin-bound thrombin, they prevent the formation of clots. They inhibit fibrinogen to fibrin, thereby limiting clot formation within the ECMO circuit without the need for binding to a cofactor like AT [74]. Only intravenous DTIs, including bivalirudin and argatroban, have been used successfully for anticoagulation in patients with ECMO.

### 8.1. Bivalirudin

Bivalirudin is a DTI potentially used as an alternative anticoagulant in patients with ECMO. Bivalirudin is approved during percutaneous coronary intervention in patients with or at risk of HIT [75]. Its use in ECMO anticoagulation is considered off-label; however, it has been used as a primary choice for anticoagulant therapy in patients with heparin resistance, HIT, or those requiring surgery [75].

Its mechanism of action is based on reversible binding to the active site of thrombin with the onset of action within 4 min after bolus administration [74]. Bivalirudin has a relatively small volume of distribution, indicating that it remains primarily within the bloodstream. It does not bind extensively to plasma proteins and is predominantly metabolised by proteolytic cleavage via plasma carboxypeptidases, forming various metabolites with reduced anticoagulant activity [76]. Bivalirudin and its metabolites are primarily eliminated renally. The half-life of bivalirudin ranges from 25 to 80 min in patients with normal renal function [77]. In patients with impaired renal function, including those with end-stage renal disease that requires hemodialysis, dosage adjustments are necessary to prevent drug accumulation and potential bleeding complications. In patients with ECMO, the reported dose of bivalirudin as a continuous infusion is in the range of 0.02–0.05 µg/kg/min [78].

### 8.2. Monitoring and Targets

The aPTT test is widely used to adjust and monitor the dosage of bivalirudin [77]. The same aPTT targets are commonly used for bivalirudin as for UFH, with the aim of a range of 1.5 to 2.5 times the patient’s baseline aPTT [78]. However, the correlation between the dosage and the response of aPTT is not flawless and has been described as weak to moderate [79]. Alternative tests may correlate strongly with bivalirudin doses, such as the ecarin chromogenic assay, the dilute thrombin time (dTT) and the chromogenic anti-IIa assay dTT, which involves plasma-diluted thrombin time and exhibits a better correlation with bivalirudin dosage compared to aPTT [80]. The bivalirudin dosing regimen and its monitoring targets are summarised in Table 1. The primary safety issue of bivalirudin lies in the absence of a specific antidote. A cessation of bivalirudin infusion can manage cases of bleeding, as its effect vanishes quickly due to its short half-life. In cases of severe bleeding, the elimination of bivalirudin can be facilitated by plasmapheresis or hemofiltration [81]; the administration of activated factor VII can reverse its effect [82]. Another issue concerns the lower effectiveness of bivalirudin under low-flow conditions with increased degradation of bivalirudin in the region of a stasis. Specifically, several studies reported increased thrombosis in reperfusion cannulas and insufficiently unloaded left ventricle in patients on ECMO anticoagulated with bivalirudin [83,84].

The limited scientific evidence supporting the use of bivalirudin in the context of ECMO anticoagulation is derived primarily from retrospective cohort studies and case series. All these studies compared patients who received bivalirudin with a control group who received UFH [85,86,87]. These reports did not describe any significant increase in bleeding or thrombotic complications associated with bivalirudin compared to UFH. Furthermore, a recent meta-analysis of nine retrospective studies showed a reduction in hospital mortality (OR = 0.65, 95% CI (0.44, 0.95), *p* = 0.03) and thrombotic events (OR = 0.55, 95% CI (0.37, 0.83), *p* = 0.004) in bivalirudin-anticoagulated patients of ECMO compared to the UFH group. However, major bleeding incidents, the duration of ECMO, and critical circuit events did not differ between the groups [88]. Based on the above reports, bivalirudin represents safe and reliable anticoagulation in patients with ECMO. However, further studies must fully elucidate its role in the ECMO anticoagulation strategy.

### 8.3. Argatroban

Argatroban represents another DTI as an alternative anticoagulant to UFH in ECMO patients. Argatroban is metabolised in the liver through several pathways, including the cytochrome P450 enzyme [89], resulting in a final half-life of 45 min [90]. There are four metabolites, one partially active [91]. Therefore, in patients with liver failure, the half-life can be prolonged up to two- to threefold, requiring adequate dose reduction [90]. Several studies report liver failure in critically ill patients, including patients with ECMO, with the need for a significant reduction in argatroban infusion rates [92,93,94,95,96]. Renal failure, with or without the need for CRRT, does not influence argatroban metabolism or clearance [95]. Therefore, dose adjustments are not necessary [97]. Argatroban is administered as a continuous intravenous infusion with reported infusion rates of 0.1–0.3 µg/kg/min in patients with ECMO [98].

### 8.4. Monitoring and Target

Similarly to bivalirudin, aPTT represents the most common monitoring method for argatroban anticoagulation, with a target of 50–70 s (Table 1) reported in the literature [98]. ACT has also been used as a single parameter or combined with aPTT to guide argatroban anticoagulation with a wide range of 150 to 230 s [98]. Furthermore, the ecarin chromogenic assay (ECA), with its high specificity to monitor DTI or VETs, did not correlate with aPPT levels [97]. Interestingly, a significant proportion of patients who reached therapeutic levels of aPTT had normal values of ECA and VETs [99]. Such findings allude to the fact that the currently used argatroban aPTT targets may not provide adequate anticoagulation in patients with ECMO. On the other hand, none of the studies reported a higher incidence of thrombotic or bleeding complications in ECMO patients anticoagulated with argatroban compared to UFH [98]. Monitoring of argatroban effect with VETs is nowadays not standardised. One study investigated VETs to guide anticoagulation in child ECMO patients with bivalirudin. The authors discovered a moderate correlation between aPTT and VET results [64].

There is no specific antidote to argatroban. In severe bleeding, activated factor VII was successfully used to reverse its effect [100]. Despite its higher cost, argatroban anticoagulation may represent an alternative strategy compared to UFH due to the reported lower administration of blood products [101].

## 9. Conclusions

The balance of hemostatic mechanisms in ECMO patients is seriously disturbed from many points of view. First, the critical condition itself is a serious factor. Second, the unique influence of a particular underlying disease must be considered, i.e., COVID-19. Third, the disturbance caused by artificial materials of the extracorporeal circuit, oxygenator, and changes in blood homeostasis also play a significant role. Last but not least, the physician’s efforts to manage possible thrombotic or bleeding events, the choice of anticoagulant agent, and the targeting of the right numbers of the monitoring modality are all critical factors that should lead to better clinical outcomes.

## Figures and Tables

**Table 1 jcm-12-06067-t001:** Summary of anticoagulant agents, dosing, and targets of the anticoagulation.

Anticoagulant Agent	Mechanism of Action	Dosing	Target
Unfractionated heparin	Inhibits mainly thrombin and Xa by activating antithrombin	Variable (start 10 IU/kg/h, titrated to reach the target) cont. inf.	ACT 180–200 s/APTT ratio 1.6–1.8/anti-Xa 0.3–0.7
Low-molecular-weight heparin	Inhibits mainly Xa by activating antithrombin	No recommendation (mainly standard prophylactic s.c. dose)	Unknown
Fondaparinux	Inhibits only Xa by activating antithrombin	No recommendation (mainly 2.5–7.5 mg/day s.c.)	Unknown
Bivalirudin	Reversibly binds to thrombin	0.02–0.05 µg/kg/min (cont. inf.)	APTT ratio 1.5–2.5
Argatroban	Reversibly binds to thrombin	0.1–0.3 µg/kg/min (cont. inf.)	APTT 50–70 s/ACT 150–230 s

cont. inf. = continuous infusion.

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
