# Peer review of "Update on Anticoagulation Strategies in Patients with ECMO—A Narrative Review"

_jcm, 2023, doi:10.3390/jcm12186067_

Round 1

Reviewer 1 Report

Line 120: please include reference

The main criticism of this work is that a specific examination of anticoagulation in the context of ECMO is not performed as stated in the title of the work. The authors perform a pharmacological examination of each single anticoagulant drug with its side effects. In my opinion, the reader is "getting lost" in all the notions of a pharmacological nature (sometimes perhaps too simple for the reader) by not fully grasping the information concerning ECMO.

In my opinion, an examination of the various settings in which ECMO is used should be carried out (I find the paragraph on COVID well done and interesting). I would perform a more specific examination of viscoelastic and thromboelastography methods which have had increasing use in many clinical settings (trauma, chronic liver disease).

In conclusion, the work risks being a pharmacological review on anticoagulants and not a specific work on ECMO

Minor editing of English language required

Author Response

Line 120: please include reference

Answer: A reference was added as required.

The main criticism of this work is that a specific examination of anticoagulation in the context of ECMO is not performed as stated in the title of the work. The authors perform a pharmacological examination of each single anticoagulant drug with its side effects. In my opinion, the reader is "getting lost" in all the notions of a pharmacological nature (sometimes perhaps too simple for the reader) by not fully grasping the information concerning ECMO. In my opinion, an examination of the various settings in which ECMO is used should be carried out (I find the paragraph on COVID well done and interesting). I would perform a more specific examination of viscoelastic and thromboelastography methods which have had increasing use in many clinical settings (trauma, chronic liver disease). In conclusion, the work risks being a pharmacological review on anticoagulants and not a specific work on ECMO.

Answer: As proposed by the reviewer, we have added more focused paragraphs on the use of anticoagulation in specific clinical scenarios including the use of viscoelastic methods (lines 177-186), differences between VA and VV setting (lines 386-399) and exhaustion of coagulation system (462-480).

Reviewer 2 Report

The authors provided an interesting and important overview of current treatment options. I would suggest including thromboelastometric methods to the discussion of every agent presented in the review. The authors should discuss anticoagulation aspects of VA and VV configuration more in detail. The authors also should discuss diagnostic indicators to distinguish between procoagulant state and exhaustion of the coagulation system.

Author Response

The authors provided an interesting and important overview of current treatment options. I would suggest including thromboelastometric methods to the discussion of every agent presented in the review. The authors should discuss anticoagulation aspects of VA and VV configuration more in detail. The authors also should discuss diagnostic indicators to distinguish between procoagulant state and exhaustion of the coagulation system.

Answer: As proposed by the reviewer, we have added more focused paragraphs on the use of anticoagulation in specific clinical scenarios including the use of viscoelastic methods (lines 177-186), differences between VA and VV setting (lines 386-399) and exhaustion of coagulation system (462-480). Also remarks on the use of viscoelastic methods were added to each antiocoagulation agent.

Round 2

Reviewer 1 Report

Despite the corrections made by the authors, the work has remained substantially the same and in my opinion lacks specificity in the treatment of the topic

Author Response

Despite the corrections made by the authors, the work has remained substantially the same and in my opinion lacks specificity in the treatment of the topic

Answer: We have made the changes and reorganization of the manuscript as requested by academic editor including the new introduction paragraph (page 1-2, lines 31-63) and LMWH in HIT paragraph (page 9, lines 412-420).